# Explanatory Model of Psychogenic, Behavioral and Environmental Causal Attributions of Cancer, and Their Psychogenic, Biomedical and Alternative Treatment in the General Population of Medellín, Colombia

**DOI:** 10.3390/bs13030236

**Published:** 2023-03-08

**Authors:** Luis Felipe Higuita-Gutiérrez, Walter Alfredo Salas-Zapata, Jaiberth Antonio Cardona-Arias

**Affiliations:** 1School of Medicine, Universidad Cooperativa de Colombia, Medellín 050010, Colombia; 2School of Microbiology, Universidad de Antioquia, Medellín 050010, Colombia

**Keywords:** cancer, health education, cancer etiology, cancer treatment, attribution theory

## Abstract

Background: Understanding the causal attributions for cancer, the elements affecting therapeutic adherence, and behaviors that may compromise people’s health or even put them at risk of dying from this disease has garnered a considerable degree of attention. Methods: This study was designed in the city of Medellín with the aim to develop and validate a model for the study of (i) the categories that can be attributable to cancer etiology, (ii) the categories that can be attributed to the efficacy of treatment, and (iii) the relationship between the categories that can be attributed to the etiology and to the efficacy of the treatment. Structural equations were performed on 611 participants. Results: The analysis revealed that attributing the disease to psychogenic factors distances people from biomedical treatments (β coefficient, −0.12), and brings them closer to psychogenic (β coefficient, 0.22) and alternative treatments (β coefficient, 0.24). Attributing cancer to behavioral factors brings people closer to psychogenic treatments (β coefficient, 0.40) over biomedical treatments (β coefficient, 0.24). Conclusions: Symbolic, cultural, and social factors were evidenced, thereby leading to the underestimation of biomedical treatments and imparting a greater degree of importance to psychogenic or alternative therapies. These therapies will subsequently affect the achievement of therapeutic objectives such as increased survival.

## 1. Introduction

Cancer causes a great impact on the health, economic, and social spheres. During 2020, 19.3 million people were diagnosed with cancer and nearly 10 million died from the disease. The global cancer burden is expected to reach 28.4 million cases in 2040 [1]. In economic terms, cancer-related costs represent 1.8% and 1.1% of the gross domestic product (GDP) in the United States and the European Union, respectively [2]. In social terms, cancer generates changes in family dynamics, transformation of roles, variation in daily life, loss of employment, and dysfunctional social relationships [3].

The clinical effects of cancer, its socioeconomic consequences, its high frequency, among other cultural, social and political factors have contributed to the development of common-sense theories related to the origin of this disease [4]. In social psychology this phenomenon is known as attribution theory, which deals with the processes through which individuals establish causal attributions to make sense of events of their life, due to a need to exercise control over them [4,5]. Causal attributions are important because they motivate and guide behaviors related to health, influence risk perception, condition measures that people take to avoid harmful practices, and determine adherence to treatments, the search for alternative therapies, and the willingness to modify health behaviors that increase the risk of becoming ill [6].

Different authors have focused on studying the causal attributions in patients with lung, breast, brain, and neck cancer, as well as up to 10 different types of cancer [5,7]. These investigations highlight three types of causal attributions: psychogenic (guilt, deserving, personality, and control of emotions), behavioral (alcohol, smoking, diet, and sun exposure), and environmental (air pollution, chemicals, toxins, and occupational risks) [5,7]. In addition, these studies show that the attributive categories of the patients are substantially different from those of the experts. For example, patients overestimate the risk of stress and environmental pollution in the etiology of their disease and underestimate the risk of behavioral factors such as obesity or sedentary lifestyle [8]. The reason for the discrepancy is that patients build and shape ideas related to their disease based on data from medical authorities and information assimilated in daily interaction with peers, influence of the cultural context, and personal experience [9,10].

Empirical studies have explored associations between the causal attributions of the disease and other health events such as quality of life, perception of disease control, and adaptation to the diagnosis. Research on the relationship between causal attributions of cancer and the attributions to its treatment efficacy is scarce. The direction (direct or inverse) and the magnitude of these relationships remain unknown. The absence of research on this topic for general population was confirmed by performing a systematic search in PubMed, OVID, ScienceDirect, Scopus, and other multidisciplinary databases with the following syntax: ((attribution [Title/Abstract]) AND (etiology cancer [Title/Abstract])) AND (treatment cancer [Title/Abstract]); Title, abstract, keywords: attribution etiology cancer treatment cancer; ((ab:(attribution)) AND (ab:(etiology cancer)) AND (ab:(treatment cancer))).

Furthermore, the majority of the research on this subject has been developed on cancer survivors, disregarding the fact that the general population represents a group of particular interest for the investigation in this field for several reasons: (i) the number of people affected: it is estimated that one in five people worldwide will develop cancer during their lifetime [1]; (ii) the influence on decision-making: previous studies have described that relatives of patients with cancer have an important role in the selection of the oncologist and hospital, and the adherence to treatment [11]; (iii) access to information: it has been documented that relatives and friends of cancer patients are the main source of information about complementary treatments and alternative medicine [12]; and (iv) secondary and tertiary prevention: beliefs related with the disease can delay the visit to the doctor after the onset of symptoms, thereby delaying timely diagnosis [13] and increasing the risk of disease progression.

Therefore, the present study was designed to assess the general population of the city of Medellín, Colombia with the aim to develop and validate a model for the study of the causal attributions of cancer, the attributions to the efficacy of cancer treatment, and the relationship between the categories attributable to the etiology of cancer and those that can be attributed to the efficacy of treatment. The general hypothesis that guided this research was the following: the psychogenic, behavioral, and environmental causal attributions of cancer have different types of relationships (in intensity and direction) with and the psychogenic, biomedical, and alternative treatment of this disease in the general population of Medellín, Colombia.

## 2. Materials and Methods

Type of study: validation of a model with structural equations.

Context: This study was developed in Medellín, Colombia. The estimated incidence of cancer in Colombia is 182 per 100,000 inhabitants and mortality is 84 per 100,000 inhabitants. Stomach, lung, breast, and prostate cancers are among the leading causes of mortality in the country [14]. Specifically, in Medellín, the second most important city in Colombia, the cancer mortality rate is higher than the national mortality average (149.6 per 100,000 inhabitants) and the figures have maintained a growing trend since 2008 [15]. In addition to the above, studies in Medellín have suggested that the knowledge, beliefs, and collective imaginary of locals with regard to cancer differ from the biomedical conceptions [16].

Population, sample, and sampling: Medellín has approximately 2,000,000 inhabitants aged >18 years, distributed into 16 communes, 6 zones, and 5 districts. We included 611 people from all areas and districts of the city to meet the Hoelter criterion (≥200 individuals to produce an adequate fit of the model) [17]. Representativeness was guaranteed using two strategies. The first strategy involved maintaining a similar proportion of individuals in the sample from each zone contributing to the size of the population; thus, the population was divided into the following zones: zone 1 northeastern (population (N) = 584,116; sample (n) = 143 [23.2%]), zone 2 northwestern (N = 510,408; n = 108 [17.7%]), zone 3 eastern center (N = 411,399; n = 101 [16.5%]), zone 4 western center (N = 357,611; n = 91 [14. 9%]), zone 5 south east (N = 110,479; n = 27 [4.4%]), zone 6 south west (N = 277,503; n = 66 [10.8%]), and districts (N = 321,704; n = 76 [12.4%]). The second strategy involved ensuring that the population sample had similar demographic and socioeconomic characteristics as follows: proportion of women (N = 52%; n = 51.1%; 95% confidence interval [CI] = 47.1–55.0), unemployed (N = 13%; n = 11.7%; 95% CI = 9.3–14.4), and self-perception of poverty (N = 30%; n = 27.2%; 95% CI = 23.8–30.8).

Measuring scales: Two surveys for the collection of information were applied, one related to the attributions to the etiology of cancer and the other related to the attributions to the treatment efficacy. The first included psychogenic (not being able to express emotions, suffering love or emotional disappointments, going through painful experiences, and harboring anger, resentment, or hate), behavioral (eating food with contaminants, sun exposure without protection, scarce consumption of fruits and vegetables, and smoking and alcohol consumption), and environmental (air pollution, exposure to radiation sources, microorganisms, and the stress of modern life) categories. The second included categories such as psychogenic treatments (avoidance of negative feelings, coping with the disease with a fighting spirit, forgiving those who hurt you, mental reprogramming, and positive thoughts and attitude), another with biomedical treatments (radiotherapy, chemotherapy, and surgery), and a last construct that includes alternative treatments (homeopathy, acupuncture, meditation, plant extracts, and animal-based beverages). All items were answered on a 4-level Likert scale ranging from “1” for those who completely disagree with a certain attribution to “4” which indicates complete agreement. Item selection and grouping in each construct was performed based on previous studies [5,18]. In addition, the appearance validity, applicability, and acceptability of the instrument were evaluated from the perspective of the subjects included and experts from the research group (doctors, oncologists, epidemiologists, medicine philosophers, and people with postgraduate degrees in social sciences).

Statistical analysis: First, the theoretical structure construct validity of the survey of causal attributions was evaluated through structural equations with the model of multiple regressions and evaluation of factorial structure with exploratory factor analysis. In this model, the items were considered as observable variables and the latent variables were the constructs of psychogenic, behavioral, and environmental etiology; the item–construct relationship was estimated using multivariate regression coefficients (adjusted for the other items of each construct) and its statistical (*p* < 0.05) and conceptual (β ≥ 0.3) significance were verified. Second, the theoretical structure of the survey of attributions to the treatment efficacy was evaluated. As in the previous model, the items were entered as observable variables and the latent variables were the constructs of psychogenic, biomedical, and alternative treatments, and item–construct relationship was analyzed using the same procedure as the first scale. Finally, the relationship among the categories (the constructs or latent variables) of the two types of attributions were evaluated with multiple regressions with multicollinearity. The regression coefficients were estimated with the maximum likelihood method.

The goodness-of-fit parameters were calculated for each model with comparative and absolute measure. In comparative fit we used the following incremental fit indices: normalized fit index (NFI), the incremental fit index (IFI), Tucker–Lewis index (TLI), and the comparative fit index (CFI), which compare the proposed model against the null, and 1 indicates perfect fit. (We considered acceptable values ≥ 0.70; we use this more lax criteria because it deals the first constructs on these oncology topics.) In the absolute measure of fit we used root mean square error of approximation per degree of freedom (RMSEA), which measures the degree to which the global model (measurement model and structural model) predicts the initial data matrix; the discrepancy between the two matrices is in terms of populations (not sample) and values ≤ 0.1 indicate good fit of the model in the population. The analyses were performed in SPSS version 27.0 and AMOS and a *p* value of 0.05 or less was consider significant.

## 3. Results

Table 1 describes the main sociodemographic characteristics of the population and the results of the items of each construct.

The explanatory model of the causal attributions to the disease showed that the items adequately represent each construct or latent variable, which accounts for the content validity of the three domains of this instrument. The lowest regression coefficients corresponded to the following items: harboring anger, resentment, or hate (psychogenic etiology construct), eating few fruits and vegetables (behavioral etiology construct), and microorganisms (environmental etiology construct). In addition, the model showed that there is a very weak relationship between the attribution of cancer to psychogenic aspects and the attribution of the disease to behavioral factors, with a β coefficient of 0.09. In turn, the model showed the presence of a strong and direct relationship between the attribution of the disease to aspects related with behavior and to aspects related with the environment, with a β coefficient of 0.86. (Figure 1)

The explanatory model of attributions to the efficacy of the treatment also showed that the items adequately represent each construct or latent variable. The lowest regression coefficients corresponded to the items of cancer removal with surgery (biomedical treatments construct) and the use of animal-based beverages (alternative treatments construct). It is important to highlight the weak relationship between the constructs of psychogenic treatments and biomedical treatments with a β coefficient of 0.01, and the strong relationship between psychogenic treatments and alternative treatments with a β coefficient of 0.83 (Figure 2).

The third model shows the relationship between the categories of the two types of attributions. In this regard, people who attributed the disease to psychogenic factors considered that psychogenic treatments (β coefficient, 0.22) and alternative treatments (β coefficient, 0.24) are effective, and biomedical treatments to be ineffective (β coefficient, −0.12). For their part, subjects who expressed a high attribution of cancer to individual behaviors considered alternative treatments to be ineffective (β coefficient, −0.32) and, instead, believed that biomedical treatments (β coefficient, 0.24) and psychogenic drugs are highly effective (β coefficient, 0.40). Lastly, those who attribute cancer to environmental factors consider psychogenic treatments as not very effective (β coefficient, −0.29) and, on the contrary, assume that alternative treatments are highly effective (β coefficient, 0.46). In this group, no significant relationship was identified with biomedical treatments (β coefficient, 0.08) (Figure 3).

## 4. Discussion

In medical sociology and health psychology there is a strong interest in the understanding of the factors related with therapeutic adherence and the risk of getting sick or dying [19]. This report contributes to this field of study by revealing the relationships between causal attributions to cancer and its treatment in the general population of Medellín. The results show that the attributions to the etiology of cancer were consistently grouped within the constructs of psychogenic, behavioral, and environmental attributions, defined in the theoretical model, similar to the model of attributions in the constructs of psychogenic, biomedical, and alternative treatments. The models of causal and treatments attributions are associated in different directions and intensity. This finding is consistent with the self-regulation model by Leventhal, who argued that when people receive a diagnosis, they begin to look for the causes of their illness and make a representation of the duration, severity, vulnerability, and consequences, based on previous personal experiences and the influence of their cultural context. This information constitutes the guide through which patients, relatives, or caregivers develop different strategies to cope with the disease, seek medical attention, choose treatments, evaluate the effectiveness of the chosen coping strategy, and strengthen or modify their choices [9,10]. I The identification of the causal attributions of a disease constitutes an important means to prevent delays in seeking medical help, lack of adherence to treatment, and help to identify factors not conceived by the health workers, which determine the effectiveness of health action [9,10].

The results of this study show that psychogenic attributions alienate people from biomedical treatments with efficacy demonstrated in controlled clinical trials, while it brings them closer to psychogenic and alternative treatments. Therefore, those who attribute cancer to behavioral factors have a high perception of the efficacy of psychogenic treatments, even above biomedical treatments. Some studies have reported that people who seek alternative and complementary treatments for cancer are more likely to reject conventional treatments such as radiotherapy and chemotherapy and have overall survival rates poorer than those who receive conventional treatments [20,21]. In psychogenic treatments, detrimental effects have been described in cancer patients since they hold themselves responsible for therapeutic failure, generating false expectations about their prognosis, leading them to avoid emotions such as fear or sadness (generating an emotional burden additional to cancer diagnosis) and are unaware of the importance of socioeconomic determinants of health that increase the risk of becoming ill or dying, or affect the possibilities of receiving treatment [22,23]. Other authors suggested that a substantial proportion of cancer patients do not reveal the use of complementary therapies to their physicians, and that physicians are generally unwilling to have discussions about complementary therapies with their patients [24]. Therefore, health professionals have an ethical responsibility to adopt a more active role in investigating the causal attributions of cancer and treatment in their patients and discuss with them the importance of adhering to conventional biomedical treatments in a timely manner because they are related to survival.

This study showed that biomedical treatments are not strongly related with any of the causal attributions studied. In fact, the relationship is null or inverse with some of them (0.08 etiology environmental and −0.12 psychogenic etiology). Patient adherence to biomedical treatment is a multicausal problem that can be affected by issues related to the patient, therapy, professionals, and health services [25]. In relation to the patients, the aforementioned attributions to the disease can be found, but also aspects such as age, education, or socioeconomic level [25]. In relation to therapy, further research is needed to contribute to the development of medications that generate fewer adverse effects and shorten the duration and the routes of administration (oral, intravenous, etc.) [25]. It is also necessary to study the place that the medical humanities, the respect for the singularity, and compassion in the care of cancer patients have in the imaginary of health professionals, since fails in recognizing these aspects can lead patients to more reluctance toward biomedical models. Regarding health services, it is necessary to investigate the existence of barriers to access to the health system in general, or cancer care in particular (costs, timeliness of care, geographic barriers) that are leading people to opt for sectors that are not biomedical.

Another important result of this study is that those who located the causes of the disease in internal or controllable factors (psychogenic and behavioral) stimulate treatments of the same order (psychogenic and alternative), which is consistent with the popularization of the idea of individual responsibility in the disease etiology and treatment. Blaming the patient for the origin of cancer and therapeutic success is an idea related to neoliberal interests that have been introduced into the culture from the ideology of positive thinking, the cult of the psyche, positive psychology, and self-help literature [26]. This idea is harmful and should be discredited for the following reasons: First, it makes states and institutions less responsible for dealing with the structural conditions that put people at risk of getting sick and receiving treatment for cancer. Second, it promotes the emergence of pseudoscientific practices such as those that state that thoughts and moods can transform the material world, attract health, and make cancer disappear [26]. Third, they disregard the broad conceptual and empirical trajectory that has been built from the field of public health around the social determinants of cancer [23].

Based on the multiple and diverse relationships between the etiological and therapeutic attributions of cancer in the general population, it is important to highlight the importance of three elements that need further research:(i)The use of scales demonstrates that the causal and therapeutic attributions of patients coexist with the biomedical model. It has been documented in other events that these attributions determine the acceptability, use, and effectiveness of the biomedical action. However, these kinds of scales are not usually considered in health care [27].(ii)There is a need to articulate biomedical health care with sociocultural factors of cancer. Factors such as risk perception, search for care, and patients’ preferences are determinants insofar as these are related to morbidity, severity, survival, and other treatment outcomes in patients with cancer [13,28,29,30].(iii)The recognition of the relationships between the causal attributions and the adherence of patients to the biomedical treatment may have some implications in patient care, treatment, clinical practice, and public health: health workers identify dimensions that affect the effectiveness of their practices and in the doctor–patient relationship it is necessary to promote person-centered medicine with recognition of sociocultural determinants of cancer; health education-communication can identify strategic issues for patients and relatives; health care must be interprofessional to impact the multidimensionality of the etiology and treatment of cancer; health systems and public health policies must be designed, implemented, and evaluated with the knowledge of the community since their attributions are determinants of the outcomes of cancer.

This work has the following limitations: (i) The causal and therapeutic attributions of cancer as a general concept were studied; however, it is necessary to remember that cancer is a concept that is used to group a set of heterogeneous diseases that originate in different parts of the body and that have different etiologies and therapeutics. In this order of ideas, it is suggested that further studies are needed that address the causal and therapeutic attributions, differentiating, at least, the most prevalent types of cancer (breast, lung, colorectal, prostate, and stomach) [1]. (ii) The models are the product of a deductive process in which the researchers defined the items and grouped them into dimensions according to the relevance described in the literature. It is important that other researchers prioritize the inductive process through qualitative-approach studies that allow the emergence of new categories of analysis to complement the results of this work. (iii) This study is cross-sectional and the causal attributions are not unchangeable; they change over time according to personal experiences, interaction with health professionals, and the influence of culture. (iv) The very fact that the attributive categories depend on the cultural context, prevents the generalization of these results in populations with cultures very different from the one that was studied in the present research. (v) Sample size, non-probabilistic sampling, and the inclusion of a single city limit the generalizability of the findings. These weaknesses need to be addressed in future studies to increase generalizability and applicability of the findings.

## 5. Conclusions

The present research demonstrated the excellent psychometric properties of the two scales. These may contribute to the advancement of research in this field, complement traditional biomedical knowledge, improve evidence-based oncology, and help to understand the sociocultural factors that determine the improvement of cancer patient outcomes.

The relationships of the psychogenic, behavioral, and environmental etiology of cancer, and its psychogenic, alternative, and biomedical treatments, were diverse in directions and intensity, and they presented simultaneously in the same subjects, which demonstrates the convergence of scientific and non-scientific elements to explain and treat cancer in the general population. These relation evidence the relevance of designing and implementing multidimensional care and treatment of cancer that allows overcoming the reductionism to the biomedical sphere, given that in the study population, symbolic, cultural, and social factors were seen to lead to a kind of underestimation or rejection of the importance of biomedical treatments with proven evidence, with a greater preponderance of psychogenic or alternative therapies, which subsequently affects the achievement of therapeutic objectives such as increased survival.

## Figures and Tables

**Figure 1 behavsci-13-00236-f001:**
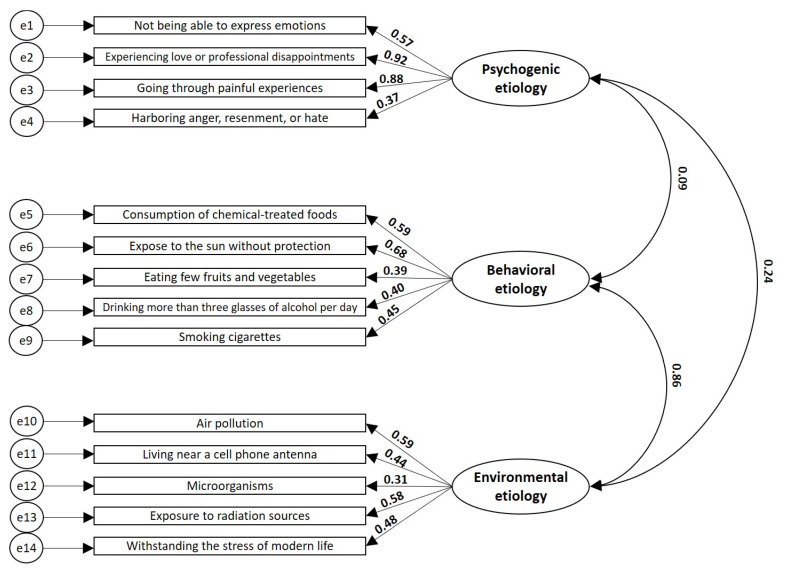
Explanatory model of the attributions to the etiology of cancer. The regression coefficients between the items (rectangles representing the observable variables) with the constructs (ovals representing the latent variables) and between the three constructs are shown. All data have *p*-values of <0.001. NFI 0.728, IFI 0.764, TLI 0.707, CFI 0.761, and RMSE 0.110.

**Figure 2 behavsci-13-00236-f002:**
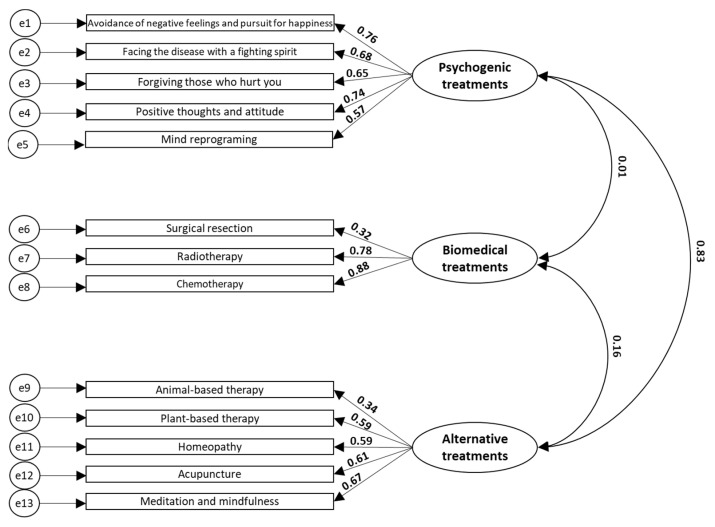
Explanatory model of the attributions to the treatment efficacy of cancer. The regression coefficients between the items (rectangles) with the constructs (ovals) and between the three constructs are shown. All data have *p*-values < 0.001. NFI 0.836, IFI 0.869, TLI 0.833, CFI 0.867, and RMSEA 0.092.

**Figure 3 behavsci-13-00236-f003:**
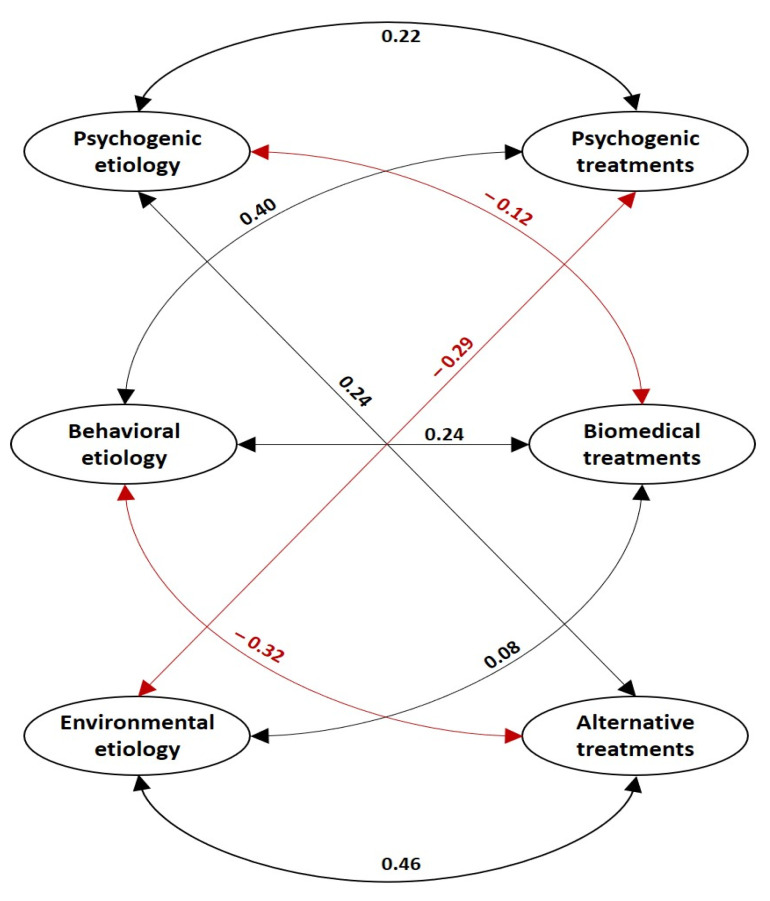
Explanatory model of the relationships between the constructs of the categories of attributions to the cancer etiology and to the treatment efficacy. The black arrows show direct (positive) relationships and the red arrows show inverse (negative) relationships. All data have *p*-values < 0.001. NFI 0.622, IFI 0.681, TLI 0.640, CFI 0.677, and RMSEA 0.093.

**Table 1 behavsci-13-00236-t001:** Sociodemographic characteristics of the population and description of items of each construct.

		n (%)
**Sociodemographic characteristics**	Gender female n (%)	312 (51.1)
	Unemployed	70 (11.7)
	Low socioeconomic status (1–2)	283 (46.3)
	Medium socioeconomic level (3–4)	289 (47.3)
		$\bar{x}\pm SD\;$ ^a^
	Age	37.0 ± 15.8
**Psychogenic etiology**	Not being able to express emotions	2.09 ± 1.14
	Suffering love or professional disappointments	2.13 ± 1.14
	Going through painful experiences	2.16 ± 1.13
	Harboring anger, resentment, or hate	2..41 ± 1.20
**Behavioral etiology**	Consumption of chemical-treated foods	3.34 ± 0.97
	Expose to the sun without protection	3.59 ± 0.83
	Eating few fruits and vegetables	2.63 ± 1.16
	Drinking more than three glasses of alcohol per day	2.65 ± 1.15
	Smoking cigarettes	3.71 ± 0.68
**Environmental etiology**	Air pollution	3.55 ± 0.81
	Living near a cell phone antenna	2.42 ± 1.16
	Microorganisms	3.29 ± 0.94
	Exposure to radiation sources	3.57 ± 0.85
	Withstanding the stress of modern life	2.89 ± 1.15
**Psychogenic treatments**	Avoidance of negative feelings and pursuit for happiness	3.01 ± 1.11
	Facing the disease with a fighting spirit	3.14 ± 1.11
	Forgiving those who hurt you	2.44 ± 1.23
	Positive thoughts and attitude	3.12 ± 1.05
	Mind reprogramming	2.33 ± 1.25
**Biomedical treatments**	Surgical resection	3.21 ± 0.99
	Radiotherapy	3.17 ± 1.12
	Chemotherapy	3.50 ± 0.87
**Alternative treatments**	Animal-based therapy	1.56 ± 0.99
	Plant-based therapy	2.85 ± 1.18
	Homeopathy	2.17 ± 1.22
	Acupuncture	1.92 ± 1.13
	Meditation and mindfulness	2.60 ± 1.20

^a^ Mean ± Standard deviation. Data presented a normal distribution according to Kolmogorov–Smirnov with Liliefors correction.

## Data Availability

Data have not been deposited in a public repository. Anonymized data are available on reasonable request to Luis Felipe Higuita Gutiérrez (e-mail: luis.higuita@campusucc.edu.co).

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
