# Peer review of "Explanatory Model of Psychogenic, Behavioral and Environmental Causal Attributions of Cancer, and Their Psychogenic, Biomedical and Alternative Treatment in the General Population of Medellín, Colombia"

_behavsci, 2023, doi:10.3390/bs13030236_

Round 1
Reviewer 1 Report
Title: Explanatory model of causal attributions and cancer treatment 2 in the general population of Medellín, Colombia.
Journal: Behavioral Sciences
Manuscript ID: behavsci-2209113
Comments
Please take care of the following points before resubmission:
Overall, scientific study and its presentation are very scientific, lucid and methodical. Needs some minir corrections and modifications.
· The manuscript is highly impressive, but the standard of English is low, frequently making it hard to understand the authors meaning and what they actually did. Language needs extensive improvement.
· Title needs modification. Make it more precise and expressive.
· Please use the correct format of the references required for the journal (check thoroughly).
· Conclusion: Make it more precise and focus on the relevance of the present study. Eliminate the irrelevant topics, which are defocusing the primary objective of this article.
Reviewer 2 Report
This manuscript, written by Dr Higuita-Gutierrez LF et al., original research, with the title of "Explanatory model of causal attributions and cancer treatment in the general population of Medellín, Colombia" performed a confirmatory factor analysis using several types of variables of phychogenic, behavior, environmental, treatments (both biomedical and alternative).
In statistics, confirmatory factor analysis (CFA) is a special form of factor analysis, most commonly used in social research. It is used to test whether measures of a construct are consistent with a researcher's understanding of the nature of that construct (or factor).
The manuscript is well written, it is easy to read, and to understand. There are enough figures and references. Both the introduction and discussion are correct.
The most important information is shown in Figures 1 to 3.
Comments:
1) Could you please expand the explanation of the methods? Could you please describe thoroughly the biostatistics?
2) How was the treatment of missing data?
3) What software was used? Did you also use AMOS?
4) Principal component, exploratory factor analysis, or confirmatory factor analysis?
5) Could you please comment on the loading factors, the standardized estimates should be at least 0.7, and many are below.
6) Regarding the model fitness, the NFI, RFI, IFI, TLI, and CFI should be >0.9.ll RMSEA <0.05, PCLOSE >0.05. etc. Values are not within the good fitting, could you please comment and explain how this would affect the interpretation of the data.
Reviewer 3 Report
Overall this is a novel paper aimed to develop and validate a model to understand the causal attributions for cancer, factors affecting therapeutic adherence, and behaviors that may compromise people's health or increase the risk of dying from this disease.
The study aimed to validate a model with structural equations for the attributions of etiology and treatment efficacy of cancer in Medellín, Colombia. The study population included 611 individuals (aged over 18 years) selected from different zones and districts of the city, to ensure representativeness in terms of demographic and socioeconomic characteristics. Two surveys were conducted to gather information on the attributions, with items answered on a 4-point Likert scale. The theoretical structure of the surveys was evaluated and the relationship between the categories of attributions was analyzed using regression coefficients estimated with maximum likelihood method.
In my opinion, the strengths of the paper are:
The study addresses a relevant topic of understanding the causal attributions for cancer and its impact on therapeutic adherence and health behaviors.
The use of structural equation modeling is appropriate for the research question and adds to the robustness of the findings.
Weaknesses of the paper are:
The sample size of 611 participants may not be representative of the general population and may limit the generalizability of the findings.
The study was conducted in a single location (Medellín), and the results may not be applicable to other populations or regions.
The study provides new insights into the impact of causal attributions for cancer on therapeutic adherence and health behaviors. However, the limitations of the sample size and location need to be addressed in future studies.
Recommendations:
Include the results of the survey in Results section as one table
The authors should provide more information on the characteristics of the sample (age, gender, education, etc.) to increase the transparency of the study.
Further studies with larger sample sizes and conducted in multiple locations are needed to confirm the findings and increase their generalizability.
The authors should provide a more detailed discussion of the implications of the results for the clinical practice and public health.
The relationships between the attributions to the etiology of cancer and the attributions to the efficacy of its treatments should be more clearly explained and discussed in the conclusion section.
The study could benefit from a more in-depth examination of the potential uses and implications of the two multidimensional scales.
The discussion section could provide more insight into the sociocultural factors that influence the improvement of cancer patient outcomes.
The manuscript could benefit from a clearer and more thorough explanation of the concept of "biologicism." (mentioned in the conclusion)
The manuscript could benefit from a more comprehensive literature review, including relevant studies on cancer attributions and beliefs in the general population.
Clarify the research question and hypothesis: The research question and hypothesis should be stated clearly at the beginning of the manuscript.
Provide more background information: Provide more background information on the topic of attributions to the etiology of cancer and the efficacy of its treatments to provide context for the study.
Discuss the role of cultural and social factors: Discuss the role of cultural and social factors in determining patient preferences and treatment outcomes in more detail.
Provide a more comprehensive discussion of the results: Provide a more comprehensive discussion of the results, including the implications of the findings for patient care and treatment.
Round 2
Reviewer 2 Report
Thank you for the answers, the revised version, and for clarifying the points.